# Key Care Provision Aspects That Affect Care Transition in the Long-Term Care Systems: Preliminary Review Findings

**DOI:** 10.3390/ijerph19116402

**Published:** 2022-05-24

**Authors:** Estera Wieczorek, Ewa Kocot, Silvia Evers, Christoph Sowada, Milena Pavlova

**Affiliations:** 1Department of Health Economics and Social Security, Institute of Public Health, Faculty of Health Sciences, Jagiellonian University Collegium Medicum, Skawińska 8, 31-066 Krakow, Poland; ewa.kocot@uj.edu.pl (E.K.); christoph.sowada@uj.edu.pl (C.S.); 2Department of Health Services Research, Care and Public Health Research Institute (CAPHRI), Faculty of Health, Medicine and Life Sciences, Maastricht University, P.O. Box 616, 6200 MD Maastricht, The Netherlands; s.evers@maastrichtuniversity.nl (S.E.); m.pavlova@maastrichtuniversity.nl (M.P.)

**Keywords:** transitional care, patient transfer, care coordination, older adults, health services research, organizing, financing

## Abstract

The aim of this brief report is to present the protocol and preliminary findings of a systematic review on key aspects of care provision that affect care transition of older adults 60+ within the long-term care systems. This brief report describes and classifies the relevant literature found in the review with the purpose to provide a base for further full systematic reviews, and to outlines a model of organizational and financing aspects that affect care transition. Our search was conducted in MEDLINE, Embase and CINAHL on 2 March 2020, before the COVID-19 pandemic. The protocol was registered at the International Prospective Register of Systematic Reviews (number: CRD42020162566). Ultimately, 229 full-text records were found eligible for further deliberation. We observed an increase in the number of publications on organizational and financial aspects of care transition since 2005. Majority of publications came from the United States, United Kingdom and Australia. In total, 213 (92%) publications discussed organizational aspects and only 16 (8%) publications were related to financial aspects. Records on organizational aspects were grouped into the following themes: communication among involved professional groups, coordination of resources, transfer of information and care responsibility of the patient, training and education of staff, e-health, education and involvement of the patient and family, social care, and opinion of patients. Publications on financial aspects were grouped into provider payment mechanisms, incentives and penalties. Overall, our search pointed out various care provision aspects being studied in the literature, which can be explored in detail in subsequent full systematic reviews focused on given aspects. We also present a model based on our preliminary findings, which enables us to better understand what kind of provision aspects affect care transition. This model can be tested and validated in subsequent research. Understating factors that affect care transition is crucial to improve the quality of transitions and ultimately the outcomes for the patients.

## 1. Introduction

Over the past decade, the concept of transitional care (i.e., actions designed to ensure safe and timely transition of patients between different levels of care) received widespread attention from clinicians, researchers, health system leaders, and policy-makers particularly [1,2]. This is due to the increasing evidence suggesting a correlation between the number of patient handovers, and medical errors or adverse events [3,4,5]. A care transition occurs when a patient moves from one care setting to another (either formal or informal care setting), and it is often a result of a change in health status or dependency [1].

### 1.1. Consequences and Reasons for Suboptimal Care Transitions

Care transitions come with a risk of negative health and quality-of-care consequences, and should be avoided or optimized when possible. Particularly, older adults with complex health issues such as chronic diseases, physical disabilities and/or cognitive impairments, and poly-pharmacy are more likely to undergo multiple transitions and are at high risk for complicated care transitions [1,6]. Poor transitions have been associated with an increase in adverse events, duplication of services, preventable readmissions to hospital, patient and provider dissatisfaction, and even increased morbidity and mortality [7]. Moreover, poor “handoff” of older patients leads to an increase in health care spending for payers, and a significant financial burden for patients [8]. Providing high-quality care and effective management of transitions are essential for good clinical outcomes and reduction of avoidable health care costs [4,7,9].

Poorly managed transitions are often a result of fragmentation of care, lack of follow-up care, confusion about medication and inadequate preparation of the patient and their caregiver for the transitional care [4,10]. Additionally, factors such as communication and information issues, inaccuracies in information exchange, and ineffective planning or coordination of care between care providers, may also compromise the quality of transitions and may result in discontinuity of care [11,12,13].

Policies and financing often focus on care in specific settings only, and neglect quality of care during transitions between the settings [1,14]. Furthermore, physicians and other clinicians tend to restrict their practices to a single setting without taking responsibility for care coordination across the continuum [15]. That is why it is crucial for governments to address the issue of transitional care by focusing on key care provision aspects as suggested by researchers [2,16,17].

### 1.2. What Is Known about Optimizing Care Transitions

Literature suggests that in order to ensure quality in transitional care, it is vital to address the care provision aspects that can influence care transition [2,16,17]. These care provision aspects can be broadly divided into organizational and financing aspects.

Regarding the organizational aspects, for example, the World Health Organization [7] argues that the organizational culture and other organizational aspects, such as communication between providers, play an important role in improving care transitions. There is a general agreement that such organizational components are vital to ensuring quality in transitional care because, currently, most professionals function in silos [16].

Similarly, regarding the financial aspects, there is an agreement that financing aspects (such as rewards and penalties) also play a significant role in care transition, as they may stimulate immediate and long-term improvements in performance [18]. Appropriate financing mechanisms are necessary for effective care transitions [19,20]. According to researchers, addressing these financial aspects of care may result in improved transitions and better care coordination [20,21].

### 1.3. Why Focusing on Care Provision Aspects as Factors of Care Transition

Although the literature indicates the possible relation between the care provision aspects and transitional care, currently, there is no review to provide an overview of these aspects. It is therefore unclear which aspects of care provision affect the care transition and could be the subject of future research. Such review could be a helpful starting point in future qualitative and quantitative studies on transitional care in a given long-term care (LTC) system. To clarify this issue, we carried out a review to gain general insight on this topic and provide recommendations for future research. We included both formal settings (care and health care institutions) and informal settings (patients’ home). Thus, the review included transitions between formal-informal, informal-formal and formal-formal settings. We excluded care provision aspects that affect transitional care within the same location. We focused the review on care for older adults 60+.

The aim of this brief report is to present the protocol and preliminary findings of our review focusing on care provision aspects that affect care transition in LTC systems. By identifying and classifying the relevant literature in this brief report, we provide a base for further full systematic reviews focused on a given aspect of care. We also use our preliminary findings to outline a model of organizational and financing aspects that affect care transition. Such a model can be a starting point in future qualitative and qualitative exploration where it can be tested and validated. This can be especially valuable for future research since such a model does not exist at the moment.

## 2. Methods

The protocol for this review has been registered in the International Prospective Register of Systematic Review (PROSPERO) under identification number CRD42020162566. The detailed protocol can be found in Appendix A. We followed PRISMA-P (Preferred Reporting Items for Systematic Reviews and Meta-Analyses) guidelines to minimize the potential bias. Below, the search strategy is briefly presented.

*Sources*: Our search was conducted in MEDLINE, Embase and CINAHL on 2 March 2020.

*Keywords*: The keywords selection and exact keywords for each database can be found in Appendix A.

*Inclusion criteria*: Studies were eligible if their focus was on transitional care between the settings among older adults 60+. Moreover, studies had to report on financial and/or organizational aspects of care transition in the LTC systems. Studies were excluded if they reported on financial and/or organizational aspects of care transition within the setting, their focus was on individuals younger than 60 years old or focused on palliative, hospice or end-of-life care. Furthermore, we included studies with primary study designs and excluded non-primary research publications.

*Selection process*: All references identified by the overall search queries were managed in Mendeley. The selection process is reported using the PRISMA chart in Appendix A. The search in the databases yielded 8342 records. After removing duplicates, 8228 publications were included in the initial screening. After reviewing the titles and abstracts, 7497 publications were excluded as they did not meet the inclusion criteria. A fraction (10%) of excluded publications was independently reviewed by a second reviewer to verify the exclusion procedure. In total, 731 publications were included for the screening based on full text. Ultimately, 229 records were included for further deliberation.

*Analysis*: Afterwards, publications were divided into: general organizational aspects, organizational disease/condition-specific aspects and financial aspects. Further details on the review and analysis are presented in Appendix A. We report here the results of the overall preliminary analysis. The results of our subsequent full systematic review on financial aspects have been recently published [22]. Other full systematic reviews can be carried out, focusing on the different organizational aspects.

## 3. Results

The literature identified in the search indicated multiple care provision aspects that may affect care transition, namely various organizational and financial aspects. Figure 1 presents a model with a classification of these organizational and financial aspects. Organizational aspects include communication among involved professional groups, transfer of information and care responsibility of the patient, coordination of resources, training and education of staff, education and involvement of the patient and family, e-health and social care. Moreover, some studies focused on financial aspects, particularly provider payment mechanisms, incentives and penalties.

Figure 2 presents the number of studies published from 2005 until 2018. As seen in the figure, the number of publications on care provision aspects, namely organizational and financial aspects, that affect care transition has been steadily increasing since 2005. Most studies identified (165 publications; 72%) have been published between 2011 and 2018. Between 2016 and 2018, the number of publications doubled, from 17 publications in 2016 to 34 publications in 2018, indicating an increased interest in care transition and care coordination. At the moment when this review was performed, there were 12 publications in 2019 and 0 publications in 2020. However, the numbers for those years might be incomplete since some publications might have still been in preparation.

Figure 3 presents the origin of publications related to care provision aspects that affect care transition. Starting from 2005, most publications come from the United States (95 publications; 41%), followed by the United Kingdom and Australia (20 publications and 18 publications, respectively). Overall, the highest number of publications were found in Northern America and Europe, while the lowest or none in Africa and Southern America (only one publication in Brazil). This may indicate that the topic of transitional care on these two continents is still not widely recognized.

Figure 4A illustrates the number of publications referring to a particular aspect of care transition. One publication could refer to more than one theme. Studies were more frequently related to organizational aspects (213 publications; 93%) than to financial aspects (16 publications; 7%). Furthermore, organizational aspects that affect care transition without special focus on any disease, were mentioned in 174 studies. Publications covered eight different themes. A high proportion, 90 (39%), of all publications on organizational aspects, discussed coordination of resources as a crucial factor that affects care transition. Particularly, nurse-led and medication reconciliation programs were of interest to researchers. Many studies also focused on the importance of transfer of information and care responsibility (51 publications), communications of involved professionals (36 publications), and education and involvement of the patient and family (18 publications). Some studies assessed the experiences and opinions regarding care transition of health professionals (22 publications) and patients and family members (17 publications). Opinions of patients and health professionals are an important source of information on factors that affect care transition.

Figure 4B presents the number of publications referring to a particular disease or health condition. In this group of publications, coordination of resources also seemed to play an important role.

The number of publications per year per category can be found in Appendix A. Between 2005 and 2019, the number of publications on care provision aspects that affect care transition increased for almost every category, indicating a growing interest in care transition and care coordination. Especially the topic of coordination of resources has been discussed in many publications for the past 15 years.

Overall, the search pointed out different care provision aspects being studied in the literature on care transition. Moreover, it identified topics that are widely investigated and themes that are under-researched. Based on those findings, we have decided to select the financing theme to develop a full systematic review study, which is already published [22]. The next full systematic review study can focus on the role of coordination of resources in care transition. Our preliminary review results reported in this brief report may already benefit other authors that intend to perform a systematic review on care transition as it provides insight into availability and scope of the publications on this topic. Furthermore, through our search, we were able to cluster factors affecting care transition into themes as presented in Figure 1. This may give an indication to other researchers and policy-makers which care provision aspects are important for care transition. Given the great variety in the publications reported above, it is important for researchers interested in reviewing the literature on transitional care to carefully consider their search strategy and particularly search string, and to narrow the scope of the study.

## 4. Conclusions

This brief report has offered the review protocol and preliminary review results on key care provision aspects, namely financial and organizational aspects, that affect care transition in LTC. The key aspect identified have also been used to create a model, which can be tested and validated in future research on the topic. This brief report is thus an initial step to gaining general insights on factors affecting care transition in the LTC systems.

As indicated by our preliminary results, in recent years, interest in care transition has grown exponentially and so did the number of publications and countries researching this topic. This is also reflected in the number of publications that have been published in the last years and the number of countries that are actively investigating this topic. Since 2005, there has been a steady growth in the number of publications regarding care transition. Increased research in this area is a response to the need to expand the evidence base demonstrating that suboptimal care transitions are quite common and are associated with worse quality of care and threaten patient safety [23,24]. In the early 2000s, most of the evidence came from the United States. Hence, the initial evidence on care transition brought attention to the problem and increased the research interest in the field of transitional care in this and other parts of the world.

These publications we identified in the review provide us with information on what are the factors that affect care transition and what are the alternative ways to optimize the care transition. Specifically, they inform policy-makers about the areas where it is important to address quality of care and patient safety in transitional care. Moreover, this area is especially important in countries where the topic of transitional care is under-researched and fundamental knowledge for future studies is lacking.

According to our preliminary results, organizational aspects of care transition seem to be more researched than financial aspects. Organizational aspects that affect care transition include: coordination of resources, communication among involved professional groups, transfer of information and care responsibility of the patient, training and education of staff, e-health, education and involvement of the patient and family, and social care. Financial aspects include: provider payment mechanisms, rewards for the role of care coordinator and penalties. Understating factors that affect care transition is crucial to improve the quality of transitions and, ultimately, the outcomes for the patients.

By the identification of different challenges and improvement measures in transitional care, it is possible to develop tailored strategies to improve clinical practice in transitional care of older adults. Our preliminary search identified that most studies on the topic refer to broadly understood care provision aspects, namely organizational and financial aspects. Authors seem to agree that these domains play a pivotal role in optimizing care transitions [16,20]. It is indisputable that good communication among involved professional groups and smooth transfer of care responsibility are crucial for optimized care transition. Professional groups should be provided with easily accessible communication channels to be able to transfer information between each other. This will help to ensure comprehensive knowledge about the patient moving from one setting to another. Good communication and transfer of information regarding health status and the needs of the patient may help the receiving setting to better accommodate patient’s needs and address preferences [7,25,26]. This is expected to have an overall positive impact on the patient’s experience of the transition process, and can also reduce poly-pharmacy and ultimately improve patient outcomes [27].

Furthermore, education of the patient and family and their involvement in the care process are as important as training and education of staff who provide care to the patient. Patient and family knowledgeable, educated and prepared for self-management and providing care at home are less likely to experience unnecessary care transitions to settings such as primary care or hospital [28]. Thus, providing education and tools for self-care and self-management enables the patient and their family to monitor and manage their disease/condition at home and avoid unnecessarily high rates of health services use and reduce costs for the health and social care systems [29]. On the other hand, providing training and education to the staff is likely to empower professionals to deliver transitional care services such as patient/family education and medication reconciliation [27]. Additionally, Bland et al. [30] found that interprofessional education increases awareness of the importance of interprofessional communication.

Another organizational aspect of care transition that should not be missed refers to the coordination of resources. Coordination of resources is an essential aspect of care transition for various reasons. For example, it is crucial not only to integrate and synchronize the activities of professionals involved in care transition, but also the availability of professionals, LTC facilities [31,32]. Specifically, a limited number of places in nursing homes and thus long waiting times may hamper the smooth transition of the patient in need of LTC. Furthermore, it is widely discussed whether eHealth and telehealth could offer promising solutions in improving communication and information exchange between the professionals, the patient and family. Various technologies could also be used to remotely monitor the biometric data of the patient or to provide remote consultations [33,34]. For instance, Pires et al. [35] reported on the potential benefit of telehealth solutions for healthcare follow-up.

Also, financial incentives may be powerful tools to stimulate the integration of care, as reported in the recent systematic review [22]. In brief, as suggested by that review, financial incentives are important drivers in improving care transition among older adults in LTC systems. Although the highest interest in financial incentives has been in primary care settings, applications of financial incentives in other settings have been reported as well, with varied impacts on care transition. Financial incentives can positively affect care coordination but not always, as studies also found unclear or no effect of financial incentives and even adverse effects [22]. Nonetheless, it is worth mentioning that studies are rather heterogeneous and results are study-specific, thus limiting comparability across countries and settings.

We acknowledge publication bias in our review because the search was carried out at the beginning of 2020, which means that it has not covered the more recent literature published during the COVID-19 pandemic. We therefore recommend a separate review to cover that period. In addition, although the screening process was checked by a second researcher, selection bias cannot be excluded. We recommend a more extensive check by a second researcher in future reviews. Also, our review was limited to care for older adults 60+, which other relevant patients groups experiencing care transition are not covered, and they can be the subject of new review studies.

Future research should also focus on a detailed analysis of a broader range of service aspects covering both provider and patient aspects of care. This can help to gain more in-depth information about alternative solutions for transitional care at the system level. In addition, future studies should focus on the implementation and feasibility of strategies to improve the care quality outcomes in transitional care among older adults in different settings and contexts.

## Figures and Tables

**Figure 1 ijerph-19-06402-f001:**
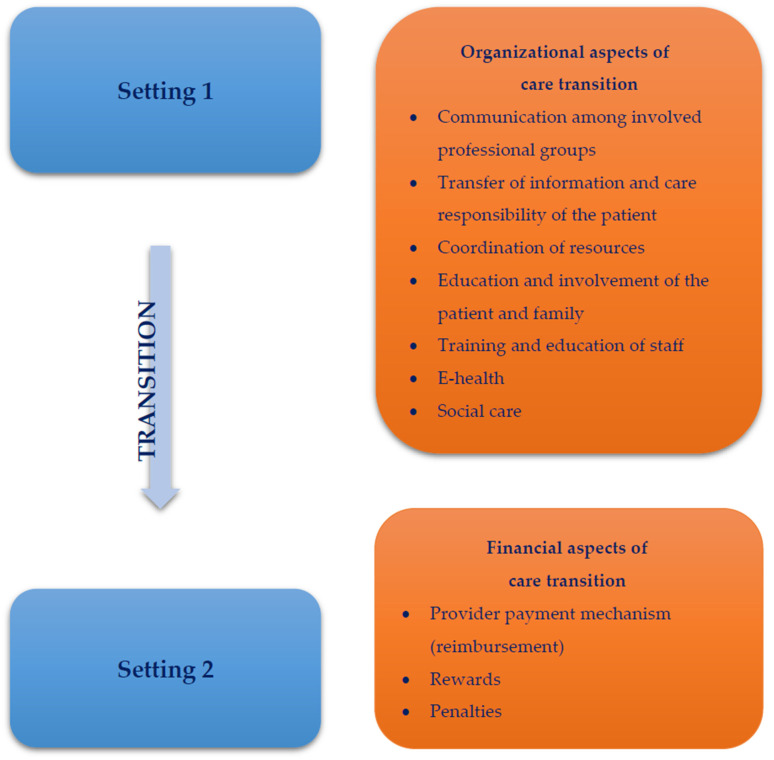
Care provision aspects that affect care transition.

**Figure 2 ijerph-19-06402-f002:**
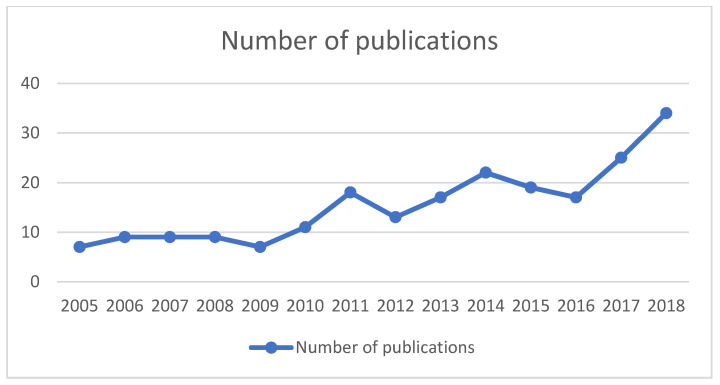
Publications from 2005 until 2018 (217 publications included) *. * Year 2019 and 2020 were excluded from the graph since the review was carried out at the beginning of 2020.

**Figure 3 ijerph-19-06402-f003:**
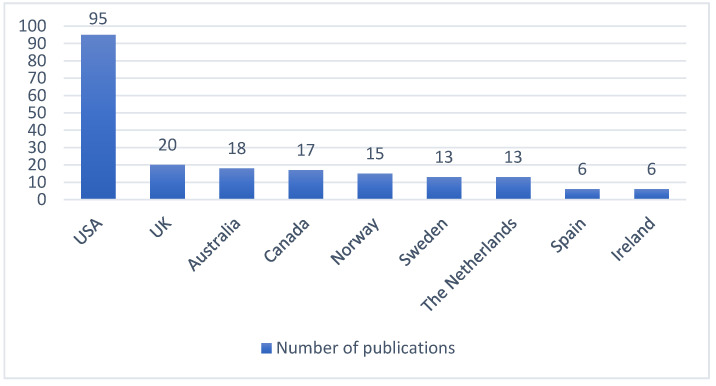
Publications by country of origin (2005–2020) (229 publications). Other countries: France (4), New Zealand (4), Taiwan (3), Germany (3), Switzerland (3), Denmark (2), Japan (2), China (Hong-Kong) (2), Belgium (1), Singapore (1), Brazil (1).

**Figure 4 ijerph-19-06402-f004:**
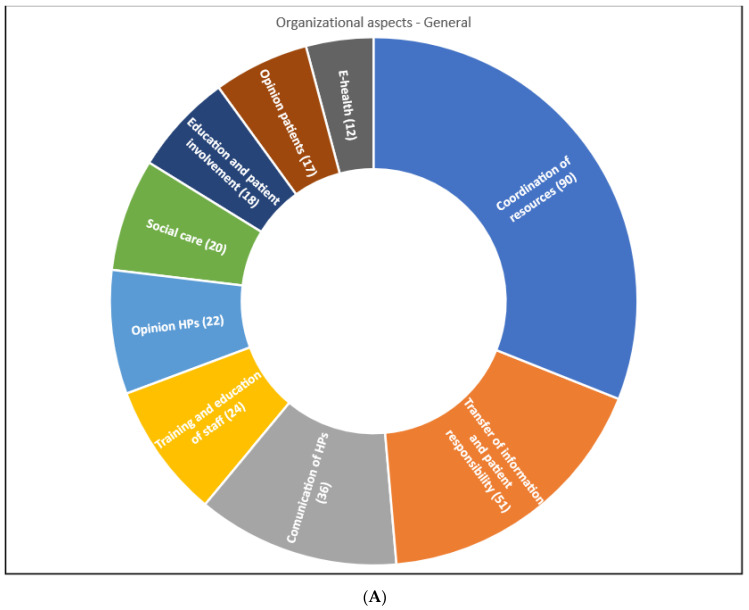
(**A**). General organizational aspects—subthemes identified in the literature. (**B**). Organizational aspects in case of a specific disease or condition—subthemes identified in the literature.

## Data Availability

The data presented in this study are available on request from the corresponding author.

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
