# Peer review of "Key Care Provision Aspects That Affect Care Transition in the Long-Term Care Systems: Preliminary Review Findings"

_ijerph, 2022, doi:10.3390/ijerph19116402_

Round 1

Reviewer 1 Report

The authors have fulfilled the comments approached by this reviewer. No further comments.

Author Response

We thank the reviewer for confirming the relevance of our revision.

Reviewer 2 Report

I am still not convinced this paper has value to readers of the journal. It is merely a description of the literature available rather than presenting the results of the literature in itself. The authors explanations do not address this issue sufficiently.

Author Response

Comment: I am still not convinced this paper has value to readers of the journal. It is merely a description of the literature available rather than presenting the results of the literature in itself. The authors explanations do not address this issue sufficiently.

Response: The specific aim of this paper is to present the protocol and preliminary findings of the review with the purpose to provide a base for further full systematic reviews and to outline a model of organizational and financing aspects that affect care transition based on the literature we reviewed. Such a model can be further tested and developed in future qualitative and qualitative exploration of the topic, which can be especially valuable since such a model does not exist at the moment.

To clarify our aim, in the revised paper, we first define the broader review aim and then explain the specific aim of this paper, which is solely focused on the protocol and preliminary findings. Given the high number of relevant papers that we identified in the review, it appeared impossible to present the full review in one single publication. Separate full reviews focused on different care aspects are needed to be able to systematically analyze the publications identified. Such separate reviews will however prevent a general overview of care aspects that affect care transition, which we are able to present here in the form of a model based on the preliminary findings. This explains the motivation of this paper.

We acknowledge that our paper is merely a description of the literature. Therefore, we have prepared and submitted it as a brief report. It is not a research article and is not intended to be a full review paper. For clarity, we now use the phrase "brief report" throughout the paper instead of "paper" or "article". As defined by the journal, brief reports are short papers (containing two figures and/or a table) that present preliminary results of a study or short complete studies or protocols. In our brief report, we combine the presentation of a review protocol and preliminary findings.

Reviewer 3 Report

Thank you so much for providing me the chance to review the revision.

The study's main weakness is that it lacks a clear purpose of the study.  The authors mentioned in their abstract "The purpose was to identify and classify the relevant literature and, in this 14 way, to provide a base for further full systematic reviews." It only gathers previous literature findings and summarizes the publication years, country of origin or publications, general organizational aspects subthemes, and specific disease or condition mentioned. The summary has no unique contribution to knowledge. They need proper methodological analysis to perform good systematic reviews and ask the appropriate questions before properly interpreting such a review and applying its conclusions to the said topic. The authors have presented the findings, but they have not interpreted or explained the findings even after the revision. 

Overall, I need to reject this paper because, as mentioned by the authors, this is only a report of their preliminary findings, so there is still a lot to be done to come up with a concrete conclusion than just providing a frequency diagram of 229 full-text records. I hope the authors will finish the review findings in order to come up with a definite conclusion and implications that will benefit the number of stakeholders.

Author Response

Comment: Thank you so much for providing me the chance to review the revision. The study's main weakness is that it lacks a clear purpose of the study. The authors mentioned in their abstract "The purpose was to identify and classify the relevant literature and, in this way, to provide a base for further full systematic reviews."

Response: To clarify this point, in the revised paper, we made a distinction between the overall aim of the review and the specific aim of this paper. The review is focused on studying the key aspects of care provision that affect care transition of older adults 60+ within the long-term care systems. However, the specific aim of this paper is to present the protocol and preliminary findings of the review with the purpose to provide a base for further full systematic reviews and to outline a model of organizational and financing aspects that affect care transition.

Comment: It only gathers previous literature findings and summarizes the publication years, country of origin or publications, general organizational aspects subthemes, and specific disease or condition mentioned. The summary has no unique contribution to knowledge. They need proper methodological analysis to perform good systematic reviews and ask the appropriate questions before properly interpreting such a review and applying its conclusions to the said topic.

Response: We acknowledge that our paper is merely a description of the literature. Therefore, our paper is prepared and submitted as a brief report. It is not a research article and is not intended to be a full review paper. For clarity, we now use the phrase "brief report" throughout the paper instead of "paper" or "article". As defined by the journal, brief reports are short papers (containing two figures and/or a table) that present preliminary results of a study or short complete studies or protocols. In our brief report, we combine the presentation of a review protocol and preliminary findings. The review protocol is checked and approved by PROSPERO, which confirms that our review is methodologically sound. Our review protocol can be especially valuable for further reviews and the model we present can be further tested and developed in future qualitative and qualitative exploration of the topic. Such a model does not exist at the moment, while it is needed to frame future studies, which explains the motivation of this paper.

Comment: The authors have presented the findings, but they have not interpreted or explained the findings even after the revision.

Response: We revised the concluding part in view of this comment and added some additional discussion of the findings reported in the results section.

Comment: Overall, I need to reject this paper because, as mentioned by the authors, this is only a report of their preliminary findings, so there is still a lot to be done to come up with a concrete conclusion than just providing a frequency diagram of 229 full-text records. I hope the authors will finish the review findings in order to come up with a definite conclusion and implications that will benefit the number of stakeholders.

Response: Given the high number of relevant papers that we identified during the review, it appeared impossible to present the full review in one single publication. We, therefore, started with an overview of preliminary findings reported here, and we continued with separate full reviews focusing on a given group of factors affecting care transition. As mentioned earlier, one of those full reviews, focused on financial aspects, has already been published. However, separate reviews will prevent a general overview of care aspects that affect care transition, which we are able to present here in the form of a model based on the preliminary findings. As mentioned earlier, such a model can be further tested and developed in future research, which indicates the relevance of this paper.

This manuscript is a resubmission of an earlier submission. The following is a list of the peer review reports and author responses from that submission.

Round 1

Reviewer 1 Report

I value the opportunity to review your manuscript - ijerph-1591692 - Key service aspects that affect care transition in the long-term care systems: Preliminary findings of a systematic literature review First, kindly check your plagiarism percentage, it is too high. 1. Introduction (1) I am not sure why you need to change the original of “Transitional Care” to “Care Transition”? (2) First, you better define what is “Transitional Care”? because in your introduction, you started providing example… (3) You are writing a literature review however; the authors are missing what is the knowledge gap from present studies that leads to the rationale of this project. (4) In 1.2, you mentioned service aspect is vital based on literature, then you follow organization aspect based on WHO, then financial aspect, so what is your main focus, and which should the reader follow from your arguments? It is like saying, all aspects are important, so? (5) In 1.3, you point out Service aspect is the most important so there is no gap from the previous literature. (6) In 1.3 also, it is not clear why the service aspect should be given priority, the reasoning is too shallow. (7) In 1.3 what do you mean by between formal-informal, informal-formal and formal-formal settings? (8) You mentioned in 1.2 that based on literature review, service aspect is vital, and then in 1.3 you mentioned that “Currently, there is no systematic review on the service aspects that affect care transition in long-term care (LTC).” So, it’s really confusing? 2. Materials (1) Kindly organize the data collection procedures, it is poorly written and it is not organize… (2) “Studies were eligible if their focus was on transitional care between the settings among older adults 60+. Moreover, studies had to report on financial and/or organizational aspects of care transition in the LTC systems. Studies were excluded if they reported on financial and/or organizational aspects of care transition within the setting, their focus was on individuals younger than 60 years old or focused on palliative, hospice or end-of-life care. Furthermore, we included studies with primary study designs 100 and excluded non-primary research articles.” (3) The collection of publications from 8342 to 731 records. You chose the paper on the basis of reading the title and the abstract only? You only started screening the full text when it goes down to 731 publications? There is a big question mark in your journal collections. 3. Results (1) You said in 1.3 that you will focus on Service aspect but why in the results you focus on “organizational and financial aspects”? I cannot follow your study, a lot of loopholes? (2) The flow of your study is disorganized, you mentioned in paragraph 2 that organizational and financial aspects are under service aspects? Really? (3) It is better to put in table form and emphasize on the difference and similarities of those paper than these figures – Figure 1, 2, 3, 4a, and 4b? You already mentioned in text and why repeat in Figure… You must prepare a Table form with different column pertaining to their findings, similarities, and differences. (4) What do you mean disease-specific aspect of care transition in relations to organizational aspect? It came out of nowhere? (5) After reading the results, there is no knowledge contribution, you just sort out all the publications and just collecting some keywords. An individual scholar can do this one using Python software. I believe that before you send these results, you need to more thorough research and provide concrete contribution to knowledge. 4. Conclusion (1) The discussion and future research are poorly written. (2) Please provide some theoretical implications and practical implications in your conclusion part. I would like to point out that I am not in the least questioning the interestingness of the results, but the scientific nature of this information, which is perhaps more useful for a report to the legal authorities or health care institute than an enrichment of the scientific literature on the subject of transitional care. Based on all of this, I think your manuscript is not publishable in its current format. I hope you find my comments helpful.

Reviewer 2 Report

Suggest dropping “literature” changing title to

Key service aspects that affect care transition in the long-term 2 care systems: Preliminary findings of a systematic re-3 view

Abstract:

I am unclear on why the preliminary findings are being presented. This is highly unusual for a systematic review and suggest salami slicing.

Line 13

Search was conducted 13 on March 02, 2020 in MEDLINE, Embase and CINAHL should read The  Search was conducted 13 on March 02, 2020 in MEDLINE, Embase and CINAHL

Line 16:

We observe an increase should read We observed an increase

Introduction

Line 58 World Health Organization [6] argues should read The World Health Organization [6] argues

Methods are well described however the review date is old – the searches were carried out at the beginning of 2020 potentially missing up to 2 years of more recent data. If this date was decided on due to eg COVID there needs to be full justification in the paper. I’m not convinced however, that this is justified given the idea of reviews is to synthesise the most up to date literature.

Results

Well written though not very exciting considering they are just about where the literature was published and the variable reported. I am still questioning why this is not just published as part of a full review.

Discussion

The authors state this review is an initial step to gain general insights on factors affecting care transition in the LTC systems – repeating my point above on the utility of this paper leading to a very limited discussion that may have more context/utility in the context of the broader systematic review.

Reviewer 3 Report

This manuscript focuses on reviewing literature in order to identify key service aspects (organizational and financing) affecting care transition. The topic is interesting and the manuscript is well-written. Nevertheless, the work is preliminar and requires more results and discussion to be appealing and worthy for a wide audience. After reviewing database and processing papers, the only contribution of this work is pointing out how many papers mention each aspect (beyond year and country). But service aspects are not properly defined and, what is more relevant, it is expected a deep disussion about why each aspect is considered influent on care transition and, if applicable, how optimizing or considering those aspects on transitions. 

The authors mention future research steps, but those should be included already here to compose an interesting and publishable work.

Minor comments:

  1. lines 104 and 109, Figure 1 is not related to the text.
  2. A sentence in Figure 1 is cut ("Education and involvement of").
  3. Review format of the text (e.g., line 70 or 147)
  4. Appendix A:
  • it includes information that should be located in the main text.
  • it refers to itself (see Appendix A) line 247
  • Does long-term care refer exclusively to elderly patients? Is other kind of patient not envisioned?
  • Table A1 is referred as Table 2 (linea 256).